# Tight Junctions, the Key Factor in Virus-Related Disease

**DOI:** 10.3390/pathogens11101200

**Published:** 2022-10-18

**Authors:** Guofei Ding, Qingyuan Shao, Haiyan Yu, Jiaqi Liu, Yingchao Li, Bin Wang, Haotian Sang, Dexin Li, Aiying Bing, Yanmeng Hou, Yihong Xiao

**Affiliations:** 1Department of Fundamental Veterinary Medicine, College of Veterinary Medicine, Shandong Agricultural University, 61 Daizong Street, Tai’an 271018, China; 2Shandong Provincial Key Laboratory of Animal Biotechnology and Disease Control and Prevention, Shandong Agricultural University, Tai’an 271018, China; 3Reproductive Center, Taian Central Hospital, Tai’an 271000, China; 4School of Basic Medicine, Shandong First Medical University and Shandong Academy of Medical Sciences, Tai’an 271016, China

**Keywords:** tight junctions, physical barriers, viruses, infection, anti-viral drugs

## Abstract

Tight junctions (TJs) are highly specialized membrane structural domains that hold cells together and form a continuous intercellular barrier in epithelial cells. TJs regulate paracellular permeability and participate in various cellular signaling pathways. As physical barriers, TJs can block viral entry into host cells; however, viruses use a variety of strategies to circumvent this barrier to facilitate their infection. This paper summarizes how viruses evade various barriers during infection by regulating the expression of TJs to facilitate their own entry into the organism causing infection, which will help to develop drugs targeting TJs to contain virus-related disease.

## 1. Introduction

Tight junctions (TJs) are primary barriers between the membranes of two adjacent cells and regulate the flux of ions and solutes, termed the “fence function” [1,2]. TJs are often found connecting epithelial or endothelial cells which line the surface of the body and body cavities by holding cells together to maintain the cellular polarity and the osmotic balance. TJs primarily consist of integrated membrane proteins: occludin (OCLN), claudins (CLDN), junction adhesion molecules (JAM) and peripheral fibrinolytic enzymes, which include ZO-1, -2, -3 [3]. CLDNs are important for the forming tight junctions, while OCLNs play a role in keeping the tight junction stable and maintaining the barrier between cells.

As an important physical barrier, TJs serve as a barrier to keep pathogen and pathogenic products outside and play a special role in the intestinal system, blood-brain barrier (BBB), blood-testis barrier (BTB) placental barrier (PB) and other natural barriers [4,5,6,7]. TJs are also the first barrier to viral infection. Both human and animal viruses usually employ variable strategies to regulate the expression, destroy or make full use of TJs to create an environment conducive to initiate their successful infection [8]. Several drugs targeted to tight junctions have been reported [9,10,11]. In this review, the mechanisms of TJs in a variety of important viruses present in humans as well as animals were summarized. How viruses hijack different members of TJs to facilitate their infection, individually, were described, which will be helpful in the development of the anti-viral drugs specificity.

## 2. TJs and Virus

As physical barriers, TJs are the first barrier to viral infection. TJs take part in the attachment step of the virus by acting as co-receptor and then help the virus enter cells. After entering the cells, viruses adopt different strategies to bypass this barrier and achieve their successful infection. Regulating the expression of specific members of TJs by virus or viral proteins is one of well-defined way to facilitate the viral infection.

### 2.1. TJs and Human Immunodeficiency Virus (HIV)

HIV is a virus that attacks the body’s immune system and causes lifelong infection resulting in acquired immunodeficiency syndrome (AIDS). HIV is transmitted primarily through direct contact with the genital fluid or blood of an infected person. The infection of HIV causes disruption of TJs function, such as the BBB and epithelial cell tight junction barrier by downregulating the expression of TJs and then increasing the permeability of TJs, so that the virus can pass freely (Table 1). Different subtypes of viral proteins have slightly different effects on the expression of TJs. CLDN 2, 3, 4, 5 and OCLN lining on vaginal barrier are usually significantly downregulated by HIV infection [12] (Figure 1). Orangutan-adapted infection causes increased permeability of the TJs by downregulating OCLN and ZO-1 in the lungs [13]. In the colon of rhesus monkeys infected by chronic SIV, mRNA expression of OCLN and CLDN3 is downregulated, and then favoring viral infection [14]. Human brain microvascular endothelial cells can also be infected with HIV, whereby the expression of CLDN-5 and ZO-1 is significantly decreased. Not only do viral particles affect TJs, but viral proteins or toxic viral products can also regulate the expression of TJs. For example, HIV Tat protein can cross the BBB [15] and is present in the central nervous system (CNS) of HIV-infected individuals. In HIV infection, Tat alters the integrity of the BBB and caused BBB dysfunction by downregulating the expression of OCLN, CLDN5, and ZO-1 [16]. HIV-1 Tat-induced downregulation of OCLN may lead to the accumulation of amyloid β in the brain, and amyloid β can severely damage brain cells [17].

TJs not only form physical and functional barriers that can antagonize microorganism, but also act as sensors of the host innate immune system [18,19]. HIV-1 infection in the endometrium and cervix induces the production of interferon regulatory factor 3 (IRF-3), which further promotes the production of interferon beta (IFN-β). IFN-β is biologically active and protects epithelial cells from HIV infection by protecting the TJ barrier (Figure 2). HIV can activate mitogen-activated protein kinase (MAPK) and then matrix metalloproteinase-9 (MMP-9) expression is upregulated [20,21]. MMP-9 interaction with tight and adherent junction proteins OCLN and E-cadherin, respectively, leading to their proteolytic degradation [22,23,24]. MAPK-associated activation of NF-κB signaling upregulate MMP-9 expression and then leads to the disruption of TJ and adhesion junctions, facilitating virus transmission between cells. HIV-1 viral particles and viral tat and gp120 proteins activate the MAPK/ERK1/2 and NF-κB signaling pathways in polarized tonsil epithelial cells, thereby upregulating the expression of MMP-9, which in turn disrupts tight junctions by cleaving CLDN1, OCLN, and ZO-1 [25]. HIV/AIDS can disrupt the intestinal epithelial barrier, mainly through viral proteins that downregulate the expression of TJs-related proteins, while the NF-κB and MAPK pathways involved interleukin-17 (IL-17) are helpful in barrier repair [26]. The expression of TJs is also altered when HIV is co-infected with other pathogens. For example, trichomoniasis, which is a common sexually transmitted infection, is associated with the transmission of HIV and preterm birth. It also induces changes in the expression of TJs, particularly the sequestering proteins, as well as the pro-inflammatory cytokines interleukin-6 (IL-6) and tumor necrosis factor-α (TNF-α) [27].

### 2.2. TJs and Flavivirus

Flaviviruses include mainly Hepatitis C virus (HCV), Zika virus (ZIKV), dengue virus (DENV), Japanese encephalitis virus (JEV), West Nile virus (WNV), and yellow fever virus (YFV) etc. The infection of most members of this family is closely related with TJs (Table 1).

#### 2.2.1. HCV

HCV causes chronic infection in at least 150 million individuals worldwide, which induces liver inflammation that can progress into advanced liver disease and eventually to cirrhosis or hepatocellular carcinoma [28]. TJs play a key role during HCV infection and multiple family members of TJs are held hostage by the virus during all stages of viral infection (Table 1). During HCV infection, TJs can be hijacked to facilitate HCV infection. Entry of HCV is primarily associated with interactions between viral and host molecules (Figure 1). The HCV envelope protein 2 (E2) of HCV complexes with the cluster of differentiation 81 (CD81) receptor and transfers to CLDN1 and OCLN, thereby facilitating viral internalization [29,30]. During CLDN1-mediated HCV infection, its N-terminal, EL1 structural domain (EL1), and 2 (EL2) important amino acid sites at the C-terminus are involved in HCV entry, and the C-terminus has an important inhibitory effect on HCV. Similarly, in OCLN, both EL2 and C-terminus 2 amino acid sites were found to be involved in HCV entry [31,32]. CLDN12 has been found to interact with the major HCV receptor, CD81 [33]. Early entry factors of CD81, the scavenger receptor B1 and the epidermal growth factor receptor co-localize on the basement membrane. HCV particles accumulate in TJs and localize to CLDN1 and OCLN in an actin-dependent manner [34]. OCLN also plays an important role in the entry of HCV into cells and its second outer loop (EC2) contributes to species selection for HCV sensitivity [35]. The first 18 residues of the cytoplasmic structural domain of OCLN can mediate HCV infection [36]. Two amino acid sites were found in the EC2 and C-termini of OCLN, involved in HCV infection [37].

Several cellular proteins were reported to take part in the regulation of the TJs. HCV infection in the context of alcoholic liver disease was associated with decreased expression of OCLN and CLDN1 in the mouse ileum and the decreased expression of TJs is closely associated with the disruption of intestinal protective factors such as intestinal trefoil factor and P-glycoprotein [38,39]. A tumor-associated calcium signaling transducer 2 (TACSTD2) was also reported to be a novel regulator of CLDN1 and OCLN. The knocking down of TACSTD2 disrupted the typical linear distribution of CLDN1 and OCLN along the cell membrane in hepatoma cells and primary human hepatocytes. In Huh7.5 cells, TACSTD2 interacts with CLDN1 and OCLN and effectively activates PKC-mediated phosphorylation of CLDN1 and OCLN, which is required for their proper cellular localization, and in order to facilitate the infection of HCV [40].

#### 2.2.2. ZIKV

ZIKV, first isolated from rhesus monkeys in Uganda in 1947, is an emerging arthropod-borne RNA virus. ZIKV is the most harmful member of the flaviviruses, which have caused millions of infections. The BBB is the boundary that separates circulating blood from extracellular fluid in the brain and CNS. The fact that ZIKV is mainly transmitted through congenital infection and causes fetal microcephaly is a strong indication that ZIKV could be transmitted freely across the PB, BTB, and BBB (Figure 1). ZIKV infection has been reported to significantly down- or up-regulate the expression of TJs, depending on the specific ZIKV subtypes. During viral infection, the expression of ZO-1 and OCLN is significantly downregulated, thereby breaking the integrity of the BBB. ZIKV infection can also disrupt cellular ZO-1 and OCLN through the proteasomal degradation pathway, resulting in PB being broken [41]. ZIKV-H subtype upregulates the expression of ZO-1 in human BBB to maintain functional barrier properties that against virus transmission. However, ZIKV-PR and ZIKV-U significantly downregulate the expression of ZO-1, OCLN, and CLDN-5 and penetrated the brain parenchyma early after infection. The disruption of TJs also increases the permeability of the BBB, which promotes viral infection and leads to severe disease of the CNS [42]. Pathological changes in the CNS are always accompanied by downregulation of TJs protein and dysfunction of the BBB. The presence of ZIKV-NS1 in the vicinity of brain microvascular endothelial cells affects the BBB, leading to microcephaly in newborns [43].

A significant reduction in CLDN4 expression was also found during ZIKV infection of the placenta of women, suggesting that the virus acts by affecting the paracellular pathway [44]. ZIKV can infect the testes of mice through sexual transmission, which further downregulates the expression of each member of the OCLN and CLDN families, resulting in significant atrophy of the seminiferous tubules and a reduction in the lumen [45,46]. ZIKV has been reported to affect the function of TJs through innate immunity, such as its ability to over activate the ERK/MAPK pathway, thereby disrupting the BTB to increase the permeability of TJs [47] and causes ZO-1 degradation via the production of inflammatory mediators, thus acting in the later stages of viral infection [48] (Figure 2).

#### 2.2.3. DENV

Most people with DENV infection are asymptomatic; only a few cases develop clinical symptoms, ranging from undifferentiated fever, to mild dengue fever infection (DF), to severe dengue fever infection (DHF). Distinct from DF, DHF is characterized by the presence of varying degrees of plasma leakage, which can progress to hypovolemic shock and circulatory collapse which is referred as dengue shock syndrome (DSS). DENV can increase the permeability between endothelial cells, leading to a disruption of the vascular system (Figure 1). In patients with DHF, this vascular disease can become severe and lead to increased permeability of a wide range of microvessels, resulting in plasma infiltration into tissues and organs [49]. In patients with DHF and DSS, the tumor necrosis factor TNF-α activates endothelial cells and promotes vascular permeability and plasma leakage. On the other hand, in human endothelial cells treated with TNF-α and DENV, the level of OCLN was significantly decreased [50]. DENV is considered one of the main causes of neurological manifestations [51]. The virus can also reach the CNS by crossing the BBB, causing severe neurological syndromes [52]. DENV can cause a significant decrease in the protein level of ZO-1 expression and peripheral localization in epithelial and endothelial cells [53].

#### 2.2.4. JEV

Japanese encephalitis caused by JEV is a neurological disease and characterized by severe pathological neuroinflammation and BBB damage. During JEV infection, the expression of OCLN, CLDN5 and ZO-1 was significantly downregulated, disrupting the integrity of the BBB [54] (Figure 1). In addition, JEV infection causes changes in inflammatory cytokine/chemokine expression. For example, IP10 induces TNF-a, which leads to BBB destruction.

### 2.3. TJs and Rabies Virus (RABV)

RABV causes a fatal encephalomyelitis that kills more than 55,000 people each year. RABV enters the peripheral neurons at the site of the wound and then enters the central nervous system through sensory and motor neurons. Enhanced BBB permeability is important during RABV infection to allow immune effectors to enter the CNS and clear RABV. Recently, it was found that BBB permeability is enhanced in mice infected with a laboratory attenuated virus, while wild-type mice showed no change. TJs also play a key role in the permeability of inflammatory cytokines caused by RABV (Figure 1). The BBB maintenance is largely dependent on interferon lambda 2 (IFN-λ2) and interferon lambda 3 (IFN-λ3), which maintains the expression of ZO-1 in the BBB thereby ensuring the integrity of TJs to reduce neuroinflammation in the Transwell model [55]. IFN-λ2 and IFN-λ3 can inhibit the replication of RABV in host cells and reduce inflammatory cytokine production in primary astrocytes and microglia, which limit the permeability of the blood-brain barrier (BBB) and prevent the excessive infiltration of inflammatory cells into the brain (Figure 2).

### 2.4. TJs and Influenza A Virus (IV)

IAV, a respiratory pathogen, causes significant morbidity and mortality worldwide and poses a serious threat to public health. Severe IAV infection can damage the alveolar epithelial cell barrier by disrupting epithelial cell tight junctions, leading to pulmonary edema and respiratory dysfunction [56]. ZO-1 has been reported to play a specific role during IAV infection. IAV infection leads to disruption of the BBB and causes influenza-associated encephalopathy through downregulating the expression of ZO-1 (Figure 1). IAV can also alter the cytoskeleton and morphology of airway epithelial cells through incomplete ZO-1 in the cytoskeleton [57]. IAV virus subtype H3N2 promotes viral infection by downregulating the expression of ZO-1, CLDN1 and OCLN when infecting ferret nasal epithelial cells [58,59]. Avian influenza virus subtype H9N2 causes intestinal mucositis injury by downregulating the expression of intestinal ZO-1, CLDN3 and OCLN [60,61] (Table 1).

Cell lines of different exhalation organs contain different TJs, e.g., primary nasal turbinate epithelial cells express CLDN-1, CLDN-3 and OCLN, porcine primary tracheal epithelial cells express CLDN-1, CLDN-3, OCLN and ZO-1 and primary alveolar epithelial cells express only CLDN-1 and CLDN-3 [62,63]. The differential expression of TJs in different tissues may have a propensity for IAV infection to be explained [64]. However, there is limited information on the mechanism, and one explanation could be that the polarity protein Lgl2 could alter the distribution of CLDN1 between cell contacts, thus weakening the barrier function of TJs and further reducing the transport of viral nucleoproteins and inhibiting their infection [65].

### 2.5. TJs and Respiratory Syncytial Virus (RSV)

RSV is a major pathogen of lower respiratory tract infections worldwide and causes asthma symptoms in children. As many as 126,000 infants are hospitalized each year in the United States for lower respiratory tract infections or bronchitis. RSV is a single-stranded, negative-stranded RNA virus belonging to the Paramyxoviridae family. RSV infection causes airway epithelial barrier dysfunction, and this barrier dysfunction may enhance the recruitment of intraepithelial dendritic cells to luminal antigens, leading to an immune response and airway inflammation enhancement [66]. In addition, RSV infection downregulates the expression of CLDN1 and OCLN, which play important roles in barrier function of bronchial epithelial cells both in vitro and in vivo [67]. When RSV infects children, large numbers of neutrophils are also being recruited into the airways of children, which significantly reduced the level of ZO-1 expression [68] (Figure 1).

### 2.6. TJs and SARS Coronavirus (SARS-CoV)

SARS-CoV is an enveloped virus that causes severe respiratory, intestinal, hepatic and neurological diseases. With the spread of the novel coronavirus SARS-CoV-2, causing very high morbidity and mortality worldwide. The SARS-CoV-2 virus displays a single-stranded RNA genome, slightly less than 30 kb in length, in which the 3’ end encodes genes for four structural proteins: spike (S), membrane (M), envelope (E), and nucleocapsid (N), of which envelope (E) is an integral membrane protein that is highly expressed in the host and is known to have an important role in coronavirus maturation, assembly, and virulence [69]. The E protein can influence the intracellular environment with the PDZ-containing protein associated with Caenorhabditis elegans lin-7 protein 1. In the pathology of SARS-CoV-infected human lungs and during the analysis of TJs in lung epithelial cells, the virus was found to affect the integrity of lung epithelial cells [70] (Figure 1 and Table 1).

### 2.7. TJs and Adeno-Associated Viruses (AAV)

AAV is a member of the genus Dependovirus which belongs in the Parvoviridae family. The prevalence of this virus in the general population ranges from 38% to 72%. AAV often requires other viruses to act together, and without this supporting virus, AAV becomes latently infected. AAV can penetrate the BBB for proliferation [71]. Crumbs 3 (Crb3) [72] and the terminal polarity determinant [73,74] are the components of the TJ complex. In Crb3 knockdown cells, the barrier composed of ZO-1 and OCLN is disrupted. AAV9 can also penetrate TJs composed of different isomers of CLDNs [75]. CLDN3, CLDN5 and CLDN11 may play important roles in AAV influencing the BTB [76,77] (Figure 1). After injecting AAV9 into the germinal tubules of mice, it penetrates both the basement membrane of the seminiferous tubules and the BTB. During this process, CLDN11 significantly inhibits the entry of the virus into the BTB [78] (Table 1).

### 2.8. TJs and Rotavirus (RV)

RV, members of the genus Rotavirus in the family Rotaviridae, are the leading cause of pediatric diarrhea worldwide, causing approximately 200,000 deaths in children under 5 years of age each year. RV primarily infect mature intestinal cells of the small intestine, but also have the potential to infect extraintestinal tissues, and infection by RVAs can be detected in cells of various epithelial origins. Infection by RV can lead to diarrhea, which is mainly caused by an increase in intestinal permeability. It has been reported that the Ras homolog gene family (RhoA)/RhoA and its downstream effector Rho kinase (ROCK)/myosin II regulatory light chain (MLC) signaling is activated in the early stages of RV infection, resulting in an early disruption of TJ integrity and changes in TJ distribution [79] (Figure 1). For example, RV infection can induce OCLN, CLND, JAM-A and ZO-1 to redistribute into the cytoplasm and increase accessory cell permeability of polarized MDCK cells. Some cellular factors were also identified to regulate TJ expression during RV infection. Glial cell-derived neurotrophic factor (GDNF) was found to be significantly upregulated in the intestinal epithelial cells of infected mice, which subsequently increased the expression of ZO-1 in infected and bystander cells. Moreover, S-nitrosoglutathione (GSNO) increased the density of the epithelial barrier by increasing ZO-1 expression [80] (Table 1).

### 2.9. TJs and Porcine Epidemic Diarrhea Virus (PEDV)

PEDV belongs to the genus Alphacoronavirus of the family Coronaviridae and is the causative agent of porcine epidemic diarrhea causing high mortality in suckling pigs. The prevalent PEDV strains are highly enteropathogenic, infecting the villous epithelium of the entire small and large intestine in the acute phase, with the jejunum and ileum being the main sites of infection. The intestinal epithelium forms a strong barrier in regulating the absorption of nutrients and water, and also restricts the entry of bacteria or viruses. In the early stages of PEDV infection, infected villi epithelial cells become necrotic and are shed, resulting in acute, severe villus atrophy. PEDV infection of epithelial cells results in the disruption of TJs and redistribution of OCLN to its intracellular location. OCLN overexpression in target cells increased PEDV infection, and the entry of virus is closely associated with internalization of OCLN [81] (Figure 1 and Table 1).

### 2.10. TJs and Porcine Reproductive and Respiratory Syndrome Virus (PRRSV)

Porcine reproductive and respiratory syndrome (PRRS) caused by PRRSV has caused huge economic losses to the global pig industry [82,83]. TJs have been reported to play an important role in the infection of PRRSV. It was found that CLDN4 expression could be downregulated in the early stages of PRRSV infection (Figure 1). The PRRSV structural protein, GP3, could interact with ECL2, the second loop of CLDN4 [9]. During PRRSV infection, CLDN4 may have a dual-functional role: (1) capture virus particles; and (2) physical barriers to cover the PRRSV cellular receptors. A model of PRRSV that makes full use of CLDN4 has been proposed. After PRRSV encounters susceptible cells, most viral particles are prevented from entering the cells through an interaction between GP3 and the ECL2 domain of CLDN4. Only parts of the viral particles bind to the cellular receptors and enter the host cells. GP3 is produced early following viral entry into the cells and ubiquitinates the transcription factor, SP1, via an unknown mechanism. SP1 was then degraded in a proteasome-mediated manner, CLDN4 transcription was downregulated, and CLDN4 expression was decreased. An increased number of PRRSV cellular receptors were exposed and a large amount of viral particles were able to enter the host cells (Table 1).

### 2.11. TJs and Porcine Circovirus Type 2 Virus (PCV2)

PCV2 is a highly prevalent worldwide and is the main causative agent of porcine circovirus-associated disease (PCVAD), a disease closely associated with post-weaning multisystemic wasting syndrome, reproductive disorders, intestinal disease and respiratory symptoms in sows. PCV2 is mainly transmitted at the respiratory level and the respiratory epithelium is an important barrier against foreign particles. PCV2 downregulates ZO-1 and OCLN in the lung and increases the permeability of the tracheal epithelial barrier to facilitate its infection [84] (Figure 1). There are several signaling pathways involved in the assembly, disassembly and maintenance of TJs in this process, including protein kinase C, Rho GTPase, myosin light chain kinase and MAPK signaling pathways. The downregulation of ZO-1 and OCLN is also due to the activating the JNK/MAPK pathway via PCV2 infection (Table 1).

### 2.12. TJs and Porcine Sapovirus (PoSaV)

PoSaV is a member of the enterovirus genus (enterovirus) of the MicroRNA family (picornaviridae), which causes severe acute gastroenteritis in pigs and which replicates mainly in the intestinal epithelium. It enters the target cells by inducing the early dissociation of TJs, which binds to the OCLN that acts as a functional coreceptor (Figure 1). The mRNA expression of ZO-1 and ZO-2, which are associated with intestinal barrier permeability, has been reported to be significantly downregulated during viral infection [85]. Moreover, PoSaV infection decreases the expression of OCLN to facilitate its entry and infection [86] (Table 1).

### 2.13. TJs and Other Pathogens

Apart from the above viruses, TJs have also been identified in several other viruses, while little knowledge of the roles the TJs in infection is available (Figure 1 and Table 1). The sexual transmission of filoviruses causes to Ebola virus (EBOV) infection, leading to the disruption of BTBs formed by intercellular TJs between testicular cells and significantly reduces the expression of ZO-1 and ZO-2 [87]. CLDN5 was decreased in the brain tissue or brain endothelial cells treated with VP1 protein of enterovirus A71 (EV-A71) and human sinusitis [88,89]. Both human parvovirus B19 (B19V) and human parvovirus (HBoV) can obviously destroy the TJs of A549 cells [90]. Infection with adult primary varicella zoster virus (VZV) is usually complicated with severe pneumonia, which eventually leads to severe acute lung injury. In a model of VZV infection, CLDN2, CLDN10 and CLDN18 expression were significantly decreased in alveolar epithelial cells [91]. Herpes simplex encephalitis (HSH) is typically caused by herpes simplex virus 1 (HSV-1) infection, and HES is also related to BBB destruction, such as ZO-1, ZO-2, and ZO-3 [92,93]. In human gastric cancer, it was found that the tumors exhibited gene fusions between CLDN18 and the small G protein regulator, kinase Rho GTPase activating protein [94]. Human parainfluenza virus type 2 (hPIV2) can cause respiratory disease when it infects airway epithelial cells. Moreover, hPIV2 can induce CLDN1 expression, which is regulated by auxin V, which regulates various host responses so that it can effectively spread the virus [95]. In the process of Hepatitis E virus (HEV) infection, CLDN5, OCLN and ZO-1 promoted HEV to invade of the CNS and destroy the BBB in Human Brain Microvascular Endothelial Cells (HBMVEC) [96].

**Table 1 pathogens-11-01200-t001:** List of tight junctions and virus.

Viruses	Viral Genome	Model/Cells	Tight Junctions	Up/Down Regulate	Effects on Virus Infection	References
HIV	ssRNA	Rhesus monkeys, Rat, HBMEC	OCLN, ZO-1, CLDN-1, 2, 3, 4, 5	↓ NF-κB and MAPK pathways involved in IL-17, IL-6, TNF-α	Inhibiting	[12,13,14,15,16,17,18,19,20,21,22,23,24,25,26]
HCV	Positive ssRNA	Mice	CLDN-1, OCLD, ZO-1	↓ Contributes to the internalization of virus, and the EC2 and C-termini of OCLN, involved in HCV infection	Promoting	[28,29,30,31,32,33,34,35,36,37,38]
HCV	Positive ssRNA	Huh7.5.1 cells	CLDN-6, 9, 12	↑ Interact with the major HCV receptor CD81	Promoting	[32,33,34]
ZIKV-H	Positive ssRNA	Mice	ZO-1	↑ Upregulates the expression of ZO-1 in human BBB	Promoting	[42,43]
ZIKV-PR	Positive ssRNA	Mice	ZO-1, OCLD, CLDN-5	↓ Downregulated the expression of ZO-1, OCLN, and CLDN-5 and penetrated into the brain parenchyma early after infection	Inhibiting	[42,43,44,45,46,47,48]
ZIKV-U	Positive ssRNA	Mice	ZO-1, OCLD, CLDN-5		Inhibiting	[42,43,44,45,46,47,48]
DENV	Positive ssRNA	human endothelial	ZO-1	↓ Downregulates the expression of TNF-α, cause a significant decrease in ZO-1 expression and peripheral localization in epithelial and endothelial cells	Inhibiting	[49,50,51,52,53]
JEV	Positive ssRNA	Mice	ZO-1, OCLN, CLDN-5	↓ Downregulation of OCLN, CLDN5 and ZO-1 expression; IP10 induces TNF-a, which leads to BBB destruction.	Inhibiting	[54]
RABV	Negative ssRNA	BHK-21	ZO-1	↓ IFN-λ is regulated by RABV, which maintains ZO-1 expression in the BBB, thus ensuring the integrity of the TJ	Inhibiting	[55]
IAV	8-segment negative ssRNA	Ferret, Mice	OCLN, ZO-1, CLDN-1,3	↓ Destructs BBB; alter the cytoskeleton and morphology of airway epithelial cells	Inhibiting	[56,57,58,59,60,61,62,63,64,65]
RSV	Negative ssRNA	Mice	OCLN, ZO-1, CLDN-1	↓ Causes airway epithelial barrier dysfunction, leading to an immune response and airway inflammation enhancement	Inhibiting	[66,67,68]
SRAS-CoV	Positive ssRNA	MDCK	ZO-1	↓ Affects the integrity of lung epithelial cells by downregulating the expression of ZO-1	Inhibiting	[69,70]
AAV	ssDNA	Mice	ZO-1, OCLN, CLDN	↓ Downregulating CLDN11	Inhibiting	[71,72,73,74,75,76,77,78]
RV	11-segment dsRNA	Humans, Mice	OCLN, CLDN, JAM-A, ZO-1	↓ Induces redistribution of OCLN, CLND, JAM-A and ZO-1 into the cytoplasm and increases accessory cell permeability of polarized MDCK cells	Inhibiting	[79,80]
PEDV	Positive ssRNA	Vero-E6, IPEC-J2	OCLN	↑ Viral entry is closely associated with internalization of OCLN	Promoting	[81]
PRRSV	Positive ssRNA	Marc-145, PAM	CLDN-4	↓ Inhibits virus adsorption, PRRSV GP3 can interact with the second loop ECL2 of CLDN4	Inhibiting	[9]
PCV2	ssDNA	Pig	ZO-1, OCLD	↓ PCV2 downregulates ZO-1 and OCLN in the lung through JNK/MAPK pathway.	Inhibiting	[84]
PoSaV	Positive ssRNA	Intestinal epithelial cells	ZO-1, 2, CLDN-1	↓ PSaV enters the target cells by inducing early dissociation of TJs, which binds to the OCLN that acts as a functional co-receptor	Inhibiting	[85,86]
EBOV	Negative ssRNA	Sertoli	ZO-1, 2	↓ Disruption of BTBs formed by intercellular TJs in testes through reduces the expression of ZO-1 and ZO-2	Inhibiting	[87]
EV-A71	Positive ssRNA	Mice	ZO-1, CLDN-5	↓ VP1 downregulates CLDN5	Inhibiting	[88,89]
B19V	ssDNA	Mice	CLDN-1, OCLN	↓ Reduces the expression of CLDN-1 and OCLN	Inhibiting	[90]
HBoV	ssDNA	Mice	CLDN-1, OCLN	↓ Reduces the expression of CLDN-1 and OCLN	Inhibiting	[90]
VZV	dsDNA	Macaques	CLDN-2, 10, 18	↓ Reduces the expression of CLDN2, CLDN10 and CLDN18 in alveolar epithelial cells	Inhibiting	[91]
hPIV2	Negative ssRNA	Airway epithelial cells	CLDN1	↑ Induces the expression of CLDN1 in airway epithelial cells	Promoting	[95]
HEV	Positive ssRNA	HBMVEC	CLDN5, OCLN and ZO-1	↓ Reduces the expression of CLDN-5, OCLN and ZO-1	Inhibiting	[96]

Notes: ss: single-stranded; ds: double-stranded. ↓: downregulated; ↑: upregulated.

## 3. Antiviral Drugs Target TJs

Due to the pleiotropic roles of TJs, several drugs or inhibitors were designed. Bowman-Birk inhibitor (BBI), a protease inhibitor derived from soybean, was recently reported to have anti-HIV-1 properties in human primary macrophages. Since an HSV-2 infection plays a role in promoting the sexual transmission of HIV-1, BBI treatment of End1/E6E7 cells can upregulate the expression of ZO-1, OCLN and CLDN5 by inhibiting ubiquitin proteasome system, and reducing the degradation of ubiquitin protein mediated by HSV-2 [97]. Peptides targeted to ECL2 domain of CLDN4 show promising application in cure PRRSV infection. This peptide can effectively inhibit PRRSV with different pathogenicity almost without any cytotoxicity. More importantly, this peptide can reduce viremia by neutralizing PRRSV particles [9].

CLDN1, as an essential HCV entry factor, is a promising antiviral target [98]. The alicyclic dipeptide MA026 is considered as an important anti-HCV drug, and MA026 has been verified as a novel irreversible TJ opener, which may be achieved by targeting CLDN1 [10]. Additionally, during the screening of HCV drugs, some studies have found that two piperazinyl benzene sulfonamides can inhibit HCV entering liver cells. These drugs mainly inhibit the viral infection by inhibiting the interaction between CD81 and CLDN1 [11,99]. Chymotrypsin can enhance BBB decomposition and the cleavage of CLDN5, ZO-1, ZO-2 and OCLN induced by JEV, as well as eliminate the decomposition of these TJs by using two different doses of chymotrypsin inhibitor TY-51469 [100]. Antibodies targeted to the first and second loops of OCLN have been showed to inhibit HCV infection in Huh7.5.1-8 cells in a dose-dependent manner without apparent cytotoxicity [101]. Similarly, houttuynia cordata treatment significantly can increase the expression of TJs (ZO-1), thus resisting influenza virus infection [102]. IL-22 treatment can upregulate ZO-1 and OCLN in HSV-2-infected cells, indicating its promising usage as anti-virus drugs [103].

## 4. Discussion

Viral disease is an important threat to human and animal fitness. TJs play an important role during viral infection. TJs are indispensable components of cell and tissue structure and involved in numerous viral infections. To help their infection, the virus takes variable strategy to regulate the expression of TJs to destroy the barriers directly or by hijacking other cellular signaling pathways. Most of studies highlight the changes in TJs proteins during viral infections, the cellular and molecular mechanisms remain to be specified. Based on the role of TJs in virus infection, several strategies are classified as: (1) destroy the integrity of TJs to enhance the permeability by downregulating TJ-associated proteins; (2) make the TJs as receptors in entering the cells; (3) ensuring the integrity of TJs to reduce neuroinflammation by upregulating TJ-associated proteins; (4) change the structure of the cytoskeleton by regulating the expression of TJ-associated proteins; and so on.

Due to their important roles in virus infection, TJs are promising targets for drug development. The present effective anti-viral drugs are developed by the monoclonal antibodies or member-specific molecules targeted to the extracellular loop domains. More and more drugs will be found, accompanying the uncovering of deep mechanism and roles in virus infection. On the other hand, TJs are important components of the mucosal barrier which modulate mucosal immune activation. This is a valuable way to fight against virus infection by enhancing the effectiveness of mucosal vaccination. Drugs targeted to keep the integrity of TJs will be a promising way to defend against virus infection.

## Figures and Tables

**Figure 1 pathogens-11-01200-f001:**
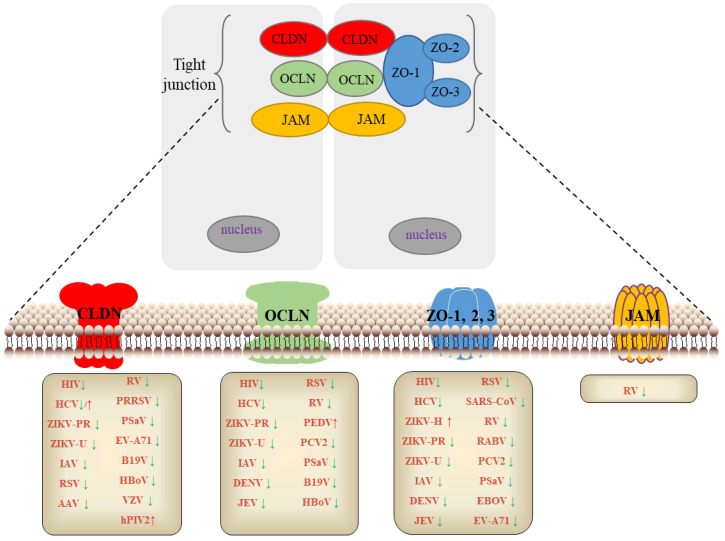
TJs expression is regulated by different viruses. The expression of TJs varies during infection by different viruses. Most viral infections downregulate the expression of TJs, but a few viruses promote their infection by upregulating the expression of TJs, such as HCV upregulates CLDN6, 9, 12, ZIKV-H upregulates ZO-1, PEDV upregulates OCLN, hPIV2 upregulates CLDN1. ↓: downregulated;↑: upregulated.

**Figure 2 pathogens-11-01200-f002:**
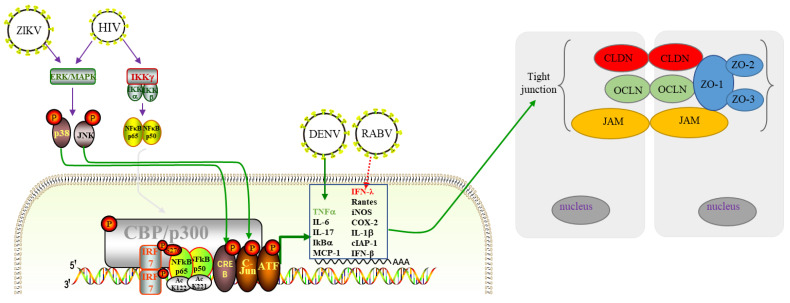
Multiple viruses regulate TJs expression through the innate immune pathway. HIV infection can activate MAPK, thereby upregulating NF-κB, and disrupting TJs, and the NF-κB and MAPK pathways involved by IL-17 participate in the repair process of the barrier. ZIKV infection can over activate the ERK/MAPK pathway and destroy BTB. Inflammatory cytokines/chemokines caused by JEV infection, such as IP-10 can regulate the expression of TJs, which is considered as the main cause of BBB destruction. Early JEV infection leads to significant upregulation of IP-10, which induces TNF-α via the JNK-c-Jun signaling pathway, and TNF-α directly contributes to BBB decomposition by inhibiting the expression of ZO-1, OCLN, and CLDN-5. DENV infection downregulates the ex-pression of TNF-α, cause a significant decrease in ZO-1 expression and peripheral localization in epithelial and endothelial cells. In the process of RABV infection, IFN-λ maintains the integrity of ZO-1 in the BBB to reduce neuroinflammation. Meanwhile, the downregulation of ZO-1 and OCLN is resulted from the activating JNK/MAPK pathway by PCV2 infection.

## Data Availability

Not applicable.

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
