# Peer review of "Tight Junctions, the Key Factor in Virus-Related Disease"

_pathogens, 2022, doi:10.3390/pathogens11101200_

Round 1
Reviewer 1 Report
The article by Ding et al. is a very interesting and well-written work on how the virus infection alters the Tight junctions (TJs). It is an area with few good references. However, as a negative point of the work is the presentation and organization of the text. For a new reader, it is difficult to follow the authors' line of reasoning. My main suggestions for improving the work are:
1. Before the detailed description of how the viruses modulated the TJs, the author could summarize how the virus infection can modulate the TJs, highlighting the processes directly affected by viral machinery, and the amplification of this process by the inflammatory response. Culminating in breaking down barriers cell by cell, to allow viruses access to other sites, favoring the expansion and spreading of the infection.
2. I am in doubt about the inclusion of HIV in the review (despite what the authors reported), it did not seem to me that TJs favored the HIV infection, unlike the other viruses reported in this paper. If I'm wrong, please make it clearer in the text.
3. I felt that biological meaning was lacking during the detailed description of how each virus modulates the expression of the tight junction proteins. I recommend that the authors express their critical opinions about these events and hypothesize the biological meaning of these processes.
Minor:
- Including the complete name of TJs in the first line of the introduction.
- Please rearrange the text of HCV and flavivirus, cause as HCV belongs to the Flaviviridae family. When HCV was presented in the text it appears that it does not correlate with flaviviruses.
Reviewer 2 Report
Dear Authors,
Thank you for providing an interesting read. At present, the review "Tight junctions, key factor regulating viral infections" provides an overview albeit brief, of the interactions between tight junctions and the different viruses. However, there are areas that has not been comprehensively covered or even discussed, as listed below:
1) The mechanisms by which the different viruses use to either cause the dissociation or dysregulation of tight junctions were only very briefly discussed upon. If the aim of the manuscript was to provide a comprehensive overview, this must be further discussed in detail. Also, what was the rationale for the selection of viruses to discuss was not clearly conveyed.
2) The possible mechanisms or mode of action for drugs targeting tight junctions were not adequately discussed and was only very briefly mentioned in the introduction (lines 40 to 41; "Several drugs targeted to tight junctions..."). This needs to be expanded and discussed in detail.
3) Tight junctions are not only limited to the main classes of occludin, claudins and JAM. There are also angulins, which have been shown to play a major role in the regulation of barrier function and is closely associated with claudin regulation. Further discussion and review on this protein and it's interaction with viruses needs to be provided within the review.
Reviewer 3 Report
Manuscript “Tight Junction the key factor regulating viral infections”. Ding et al., explains the role of Tight Junction proteins in regulating viral infections. Authors have covered most of the infectious virus with the available data, but I found that this manuscript needs more information and clarifications, especially there is lot of mismatches between the text and the cited references.
First of all, it is difficult to say that TJ as a key factor in regulating viral infections, a comprehensive study data still not available on TJs and their role in several viruses. In addition, there are very limited studies on TJs in some viruses. Thus, it is too early to say that these are key factors in viral infections, authors may modify their title of the manuscript to reflect it. Moreover, TJs function in different infectious viruses is different. Authors have misinterpreted lot of references that leads to confusions. The authors are requested to go through the references cited more carefully and confirm their data presentation is accurate.
Line 58-59 need modification- is there any direct evidence that TJs were regulated by viral proteins, if so, provide more information.
Line 74-79, More clarification needed, how does the upregulationof MAPK – NFkB axis through MMP9 activate TJs, can give more details of the pathway or explain the mechanism of action here.
Fig 2: line 90: Please be more specific on the role of IP10, does it up regulate or down regulate the expression of TJs?.
Line 125, explain the role of phosphorylation of OCLN and CLDN1 and viral infections.
Line 139, can you give more details on which TJs are affected.
Line 175-177 redundant-delete it.
Line 174, does the decrease in ZO-1 expression, is that due to degradation of the protein, or the gene expression?
Line 182, need modification- IP10 induces TNFa and that leads to BBB destruction.
Line 192, can you be more specific on the inflammatory cytokine, which cytokines?
Line: 208, Interpreted differently in case of reference #54, which shows studies in ferret nasal epithelial cells and not human. The authors are requested to more carefully go through the references and the accuracy of data is very essential.
Lines 211-214, provide references to the statements of each cell.
Line215-218: Confusing, If LgI2alter CLDN1 and weakening barrier function, then how it will it reduce the viral nucleoprotein transportation, it could increase as the barrier breaks, clarify?
Line 229-230, reference required for neutrophil recruitment.
Reference 61, study in a murine model and not in pediatric patients, correct it.
Line Ref #63, results misinterpreted, paper says “Conversely, the tight junction protein ZO-1 was maintained at cell–cell contact domains in SARS-CoV–infected cells, indicating the specificity of PALS1 mislocation” not mentioning on the role of ZO1 downregulating leads to the TJ integrity, instead it’s the PALS1 plays a determinant role in the disruption of the lung epithelium in SARS patients.
The discussion part needs additional points, how does same TJs (such as ZO-1), interact or how they differ in function with different viruses. Does the same TJ protein differ in function in different cell lines or in tissues and can we come up with a generalized model for a group of viruses (single stranded vs double stranded, RNA viruses vs DNA viruses etc.).
Reviewer 4 Report
This review describing the role of Tight junctions and how they regulate t viral infections by Ding et al., is overall a well written review. It covers a lot of bases and is relevant to the field.
Author Response
This review describing the role of Tight junctions and how they regulate t viral infections by Ding et al., is overall a well written review. It covers a lot of bases and is relevant to the field.
Responses: I am very grateful to the comment.
Round 2
Reviewer 1 Report
I believe the article is now suitable for publication.
Author Response
Responses: I am very grateful for your comment on the revision of our paper.
Reviewer 2 Report
Dear Authors,
Thank you for taking the time to address the comments. The revised manuscript now reads better and provides slightly more clarity. However, it still requires more details or discussion areas in order to stand out from other reviews which had previously covered similar topics (Linfield et al, 2021, Airway Tight Junctions as targets of viral infections). Inclusion of these areas will then value-add to the overall manuscript to distinguish it from other similar reviews.
Reviewer 3 Report
Authors made significant modifications and additions to the text and references.
Author Response
Responses: I am very grateful for your comment on the revision of our paper.
We have gone through the whole paper for English changes.